# Exploring Resilience in Care Home Nurses: An Online Survey

**DOI:** 10.3390/healthcare11243120

**Published:** 2023-12-08

**Authors:** Anita Mallon, Gary Mitchell, Gillian Carter, Derek Francis McLaughlin, Mark Linden, Christine Brown Wilson

**Affiliations:** School of Nursing & Midwifery, Queen’s University Belfast, 97 Lisburn Road, Belfast BT9 7BL, UK; gary.mitchell@qub.ac.uk (G.M.); g.carter@qub.ac.uk (G.C.); derek.mclaughlin@qub.ac.uk (D.F.M.); m.linden@qub.ac.uk (M.L.)

**Keywords:** resilience, care homes, nurses, social-ecological theory

## Abstract

Resilience is considered a core capability for nurses in managing workplace challenges and adversity. The COVID-19 pandemic has brought care homes into the public consciousness; yet, little is known about the resilience of care home nurses and the attributes required to positively adapt in a job where pressure lies with individuals to affect whole systems. To address this gap, an online survey was undertaken to explore the levels of resilience and potential influencing factors in a sample of care home nurses in Northern Ireland between January and April 2022. The survey included the Connor–Davidson Resilience Scale, demographic questions and items relating to nursing practice and care home characteristics. Mean differences and key predictors of higher resilience were explored through statistical analysis. A moderate level of resilience was reported among the participants (n = 56). The key predictors of increased resilience were older age and higher levels of education. The pandemic has exposed systemic weakness but also the strengths and untapped potential of the care home sector. By linking the individual, family, community and organisation, care home nurses may have developed unique attributes, which could be explored and nurtured. With tailored support, which capitalises on assets, they can influence a much needed culture change, which ensures the contribution of this sector to society is recognised and valued.

## 1. Introduction

The Sars 2 Coronavirus (COVID-19) pandemic has exposed pre-existing systemic weaknesses in the structure and functioning of health and social care systems across Europe [1]. These systems’ failures have enabled the marginalisation of care homes for older adults, placing them not only outside of health care priorities but also outside of public consciousness [1,2,3]. However, the pandemic has also increased the visibility of care homes and highlighted their centrality to a functioning health and social care system [2]. Within care homes, nurses not only manage the delivery of care for the organisation, but they also have a pivotal role in providing the link between the individual resident, the family, the community and the multi-disciplinary team. The job demands on the care home nurse are significant; they are often the only registered professional on duty leading a team caring for residents, who may have multiple complex health conditions [4]. These demands can lead to emotional exhaustion and burnout [5]. Despite the efforts made to protect the residents, staff in care homes have reported how the pandemic has reinforced a feeling of reduced value compared to NHS colleagues [6,7].

Feeling valued and respected is associated with care home work related wellbeing [3] and can act as a buffer to psychological distress [3,8,9]. Advocacy for individually tailored supportive programmes aimed at care home nurses has followed qualitative inquiry into the burdens of care and the emotional welfare of care home nurses [10,11]. Supporting the resilience and wellbeing of care home nurses should be a public health priority, requiring an all-systems approach, which values, protects and empowers the professionals who provide a vital societal service. Yet, little is known about the levels of resilience of care home nurses, what unique attributes they bring to the work, what can be nurtured, supported and what can be learned.

The pandemic has exacerbated the levels of psychological distress among health care workers, with nurses identified as a particularly at-risk group [12,13,14]. Nurses, by the nature of their work, face a myriad of challenges; yet, some not only recover but thrive in the face of adversity [15]. Care home nurses have reported their role as challenging, multi-faceted, with high expectations felt throughout the system [4]. As leaders and advocates, care home nurses are responsible for maintaining the safety, dignity and personhood of residents [4,16]. This has been challenged through the pandemic, with care home staff relating difficulties in finding a balance between protecting the residents from infection and enabling a degree of self-determination [6]. Being flexible and positively adapting to challenges is a core capability of care home nurses, as they provide care for people with varying levels of frailty, cognitive impairment and co-morbidities [17] while managing a high staff turnover [18,19]. Isolation, illness and fear relating to the pandemic have intensified these challenges, with detrimental health implications for some staff [20,21]. In a study of resilience, stress and wellbeing among health care employees (n = 1130) [22], nurses (n = 136) were reported as having the highest level of distress, as scored by the short, validated wellbeing index [23]. A significant negative effect on the psychological wellbeing and quality of working life of nurses, midwives and allied health professionals was evidenced in a recent study measuring coping, wellbeing and burnout at two time points of the COVID-19 pandemic [24]. Resilience can help mediate some of these workplace stresses; it is seen as protective and is enhanced by many internal traits and external influences [25,26,27]. The concept of resilience may go some way to explain the sense of professional satisfaction, dedication and resourcefulness to overcome barriers to care, which has been demonstrated throughout the sector [6,28].

Fundamental to resilience is the presence of adversity as an antecedent and positive adaptation as a consequence [29]. Resilience differs from hardiness and ‘just coping’ by focusing on strengths within the individual and the environment, which enable positive adaptation when faced with significant adversity [30,31]. While the trait aspect of resilience has been recognised [32], the dynamic nature of resilience has gained increasing traction as something, which can be changed and will change over the life span [33,34].

A concept analysis of resilience as it applies to nursing defines resilience as the ability to adapt and avoid psychological harm while providing optimum care [35]. While studies have sought to identify the factors contributing to individual resilience, recent reviews of evidence relating to resilience in nursing report the influence of socio-demographic factors—such as age, marital status, dependents, education, nursing experience and job grade—as conflicting [25,36]. In a demographic factor synthesis undertaken of 14 studies, which investigated nurse resilience, Yu [25] concluded there were inconsistencies with regard to whether socio-demographic factors were associated with resilience. For example, positive correlations between participant age and resilience scores have been reported in studies measuring nurse resilience [37,38]. However, in other studies, age was found not to have a significant relationship with resilience [39,40]. While education was seen to positively correlate with resilience in some studies prior to the pandemic [41,42], a recent review of the factors influencing resilience during the COVID-19 outbreak found that higher education was associated with non-resilient outcomes [43]. Again, clinical rank has been significantly correlated with resilience (*p* < 0.05) in a survey of hospital nurses (n = 2981) in China [44], where those with higher clinical rank were found to be more resilient, but no association was found in an exploratory study in another region of China (n = 1356) [41]. Studies relating to marital status are equally inconsistent, with one study comparing the resilience of nurses in Singapore identifying a strong association with resilience (*p* < 0.001) [44], while another study found no significance (*p* = 0.457) [45]. Difficulties with sampling, defining exactly what is being measured and the context may explain some of the variability in the contribution of these variables to resilience. However, there is a dearth of evidence as to which socio-demographic factors influence the resilience of care home nurses.

An integrative review addressing resilience in nursing undertaken in 2021 identified two studies, which focused on care home nurses. The first study was a qualitative study focusing on the factors impacting resilience [46]. This study identified the importance of meaningful relationships with residents as enhancing resilience and the importance of social support opportunities to debrief and connect. The second study measured resilience against organisational empowerment and self-perceived quality of care. It reported resilience as being positively associated with higher perceived quality of care [47]. In a study of Australian nurses (n = 1743) from across private and statutory care, resilience was found to be a key variable impacting compassion satisfaction [48]. However, these studies were undertaken prior to the pandemic; the isolation measures introduced since have been perceived as negatively impacting the connection nurses have with residents and communities [6]. A small number of studies among care home nurses have been undertaken in the last two years, revealing negative outcomes related to low resilience, such as exhaustion, burnout and stress, but also resilient outcomes in terms of forging different connections and innovation [6,49,50,51].

Studies highlight that nurse resilience can protect against psychological harm, and higher resilience can predict increased levels of happiness, wellbeing, compassion, satisfaction, and it may increase job satisfaction and retention [52,53,54]. Reviews of empirical data suggest that resilience can act as a buffer against burnout, secondary traumatic stress [52,55], exhaustion, anxiety and depression [56]. In a study of psychological distress and resilience levels among health care workers in geriatric services in Italy during the pandemic (n = 818) [14], high levels of resilience accounted for 20% of the variance in the GHQ-12 General Health Questionnaire [57], with high levels of resilience predicting lower levels of distress (β = −0.25, *p* < 0.001). While there has been more focus on nurse resilience in recent years, with studies involving hospital nurses in general and specialist units [44,58,59,60], there remains a dearth of research on supporting the resilience of care home nurses [61].

Reviews of interventions aiming to support nurse resilience highlight three main points: (1) the need for an unequivocal understanding of resilience, (2) that resilience should be measured as a primary outcome, and (3) investigation of resilience should extend beyond the general hospital, in particular to workers in social care [26,62,63,64]. This study sought to determine the levels of resilience in care home nurses in one region of the United Kingdom and to explore the factors which may predict higher resilience. We hypothesised that there will be a statistically significant relationship between resilience and key demographics, namely age, education, marital status, experience and clinical grade. To test these hypotheses, we conducted a survey to determine the levels of resilience and to explore the effect of these variables on resilience scores.

## 2. Materials and Methods

This paper reports the findings of a cross-sectional online survey exploring the resilience of registered nurses working in care homes in Northern Ireland. The guidelines for reporting cross-sectional studies (STROBE) were used [65]. This is the first phase of an explanatory mixed-methods study developing a co-designed e-resource with care home nurses to foster resilience. The study is underpinned by the socio-ecological theory [66], recognising the multiple systemic influences on individual resilience, as highlighted in the literature. A socio-ecological perspective explores the interrelations between personal, social and environmental factors, which impact health and illness [67]. The resilience framework proposed by Windle and Bennett [68] highlights three areas of influence, namely individual, community and societal. The perspective of this study frames the research around a layered model, as that proposed by socio-ecological theorists Simons-Morten, McLeroy and Wendel [69], where the organisation within which the individual works, the policies, rules and regulations governing the individual and the organisation and the societal culture may all potentially impact the resilience of care home nurses.

This paper explores variables primarily at the individual, community and organisational levels. At the intrapersonal level, the data considered the age, gender, years of experience, experience during COVID-19, education, ethnicity, clinical grade, shift work and employment practice. At the interpersonal level, data relating to marital status and dependants were collected. At a community level, the location of the home was identified as rural or urban. At the organisational level, care home capacity and geographical placement were collated; these variables incorporate the context of the care home, a factor highlighted as central to resilience [70].

### 2.1. Research Setting and Sample

The term ‘care home’ is used as a broad term to represent any residential facility for older people. The sample and inclusion criteria for this pilot study were (1) registered nurses (2) presently employed in care homes in Northern Ireland. A convenience sample of nurses were recruited using care home support groups and professional networks of the research team. Through these networks, the managers of the care homes were contacted via an invitation email containing a link to the participant information sheet and survey and asked to share the survey with registered nurses working in their care homes. The consent to participate was obtained prior to accessing the questions in the survey. The diversity of this population in terms of geography and organisational context meant that several strategies were used to boost recruitment; a reminder email was sent after 2 weeks, and a prize draw for a voucher worth GBP 50.00 was offered to participants who completed the study. The university social media platforms were used to advertise the study, and a flyer with a QR code directly to the participant information sheet and survey was sent with the invites. Data collection took place between January and April 2022.

### 2.2. Ethical Considerations

The anonymous online survey was reviewed and granted ethical approval by Queen’s University Belfast, Medical Health & Life Sciences Ethics Committee, on 17 December 2021 (Ref MHLS 21_137). The participant information sheet contained detailed information on the rights and obligations of participants and data storage in line with the Data Protection Act (2018). Participants were asked to confirm they had read the participant information sheet and consented to participate in the study prior to completion.

### 2.3. Measures

This study followed a quantitative, descriptive approach using a cross-sectional survey design to gather data. The survey included the 25-item Connor–Davidson Resilience Scale (CD-RISC-25). A further fourteen items relating to demographics, the nursing role and characteristics of the care home—identified through the literature as potentially influencing resilience levels—were also collected. The Connor–Davidson Resilience Scale [31] was employed to measure the levels of resilience. The choice of scale was informed by the literature and a methodological analysis of the resilience scales, which found the CD-RISC scale in the top three scales due to its psychometric properties [71]. The scale was developed in general and clinical populations, with samples demonstrating internal consistency with a Cronbach coefficient alpha of 0.89 in the general population and good test–retest reliability in the clinical population [31]. In this study, the Cronbach coefficient alpha was 0.92, indicating excellent internal consistency. The CD-RISC has been used consistently to measure resilience among diverse groups of the population [72]. It is the most frequent measure of resilience across studies measuring resilience in health care workers [62,73], and specifically among nurses [25,36,74].

The 25 statements are scored on a 5-point Likert scale, from ‘not true at all’ to ‘true nearly all of the time’, and the scores range from 0 to 4. The total score is obtained by adding up the scores of the 25 items. Higher scores indicate greater resilience, and lower scores denote less resilience. The scale relates to how the respondents have felt in the last month, and the total scores range from 1 to 100. The factor analysis performed on population data revealed five factors: (1) personal competence, high standards and tenacity; (2) emotional and cognitive control under pressure; (3) positive acceptance of change and secure relationships; (4) control; and (5) spiritual dimension. The factor structure has been reported as being different in differing populations, varying from three to five factors [31,75]; however, a study of hospital nurses revealed a five-factor solution in keeping with the original scale analysis [39]. The authors of the scale were contacted and provided permission to use the scale.

### 2.4. Data Analysis

Data were held in an online repository (www.qualtrics.com) accessed 15 November 2021 and exported to SPSS Version 28 for statistical analysis. Missing data were addressed prior to the analyses. The data from 16 participants were removed, as over 80% of the data was missing, which left 56 participants included in the final analysis. Using descriptive statistics, the characteristics of the participants and data relating to the nursing role and experience and the context of the care home were gathered. A total resilience score was also computed from the CD-RISC. The total score was examined for normality of distribution and was shown to have a normal distribution, thus allowing for parametric testing. Bivariate analysis was undertaken using Pearson’s correlation, independent *t*-tests and one-way analysis of variance (ANOVA). Five predictor variables, namely age, education, years of experience, job grade and marital status, were then entered into a linear regression model to determine their strength of association with resilience, as measured by the CD-RISC. The inferential tests were two tailed, with significance set at *p* ≤ 0.05. As there is a dearth of literature examining which socio-demographic factors influence resilience, the variables were not entered in any order. Linear regression was used to explore the predictive value of age (continuous variable), years of experience (continuous), clinical grade (manager/nurse), education (certificate or diploma/degree+) and marital status (married, partnership/single) on resilience scores. Examination of the correlation matrix revealed a correlation of >0.8 between age and years since registration; therefore, years since registration was removed from the final model. The results of the linear regression guided further inferential testing relating to which individual factors (see Section 2.3) could be associated with the predictor variables.

## 3. Results

In this section, we describe the characteristics of the population and identify the mean resilience scores and significance across the participant demographics. Linear regression is used to identify the potential predictors of resilience. Inferential testing on the scale factors is undertaken to explore the possible associations.

### 3.1. Demographic, Nursing and Care Home Characteristics

In total, 72 responses were received, with 56 valid datasets. The majority of respondents were female (n = 50, 89%) and of white ethnicity (n = 48, 86%). The mean age was 46.4 years (SD = 9.5 years, range 27–60 years). Most respondents were married or in a domestic partnership (n = 45, 80%). The mean years since registration was 22.5 years (SD = 12, range 1–44 years). The majority of participants worked daytime shifts (n = 46, 82%) and worked on a full-time basis. The capacity of the homes in which they worked varied, with 90% of the homes having a capacity of <80 residents. Homes in rural areas accounted for 25% (n = 14), and homes in urban areas represented 75% (n = 42) of survey respondents. In total, 95% of participants had worked in excess of 12 months during the pandemic, with only three participants who were novice nurses having worked under a year in the care home. Nurse Managers formed 66% (n = 37) of the sample, with nurses and one sister accounting for the other 34% (n = 19). Table 1 presents the descriptive characteristics of the sample.

### 3.2. Resilience

The mean score for resilience, as measured by the CD-RISC, was 77.36 (SD = 11.31, range 52–100).

### 3.3. Resilience Scores

The tables below (Table 2 and Table 3) present the descriptive results showing the distribution of mean resilience scores relating to socio-demographics and nursing and care home characteristics. Some differences in scores were apparent in the descriptive analysis, such as higher scores for those who were older and those who were single. Participants who had dependants also scored higher. However, a comparison of the means via independent t-tests and ANOVA reported no differences, which were significant at the *p* ≤ 0.05 level.

Higher resilience scores were recorded for those participants who worked less than 12 months during the pandemic. However, this applied to only three participants, who were newly registered nurses. There was a general increase in resilience scores with years of experience. These differences did not reach significance at the *p* ≤ 0.05 level.

### 3.4. Multiple Linear Regression

Multiple linear regression was used to assess the ability of the five independent variables to predict resilience scores. Years since registration was removed from the model due to a high correlation with age, at 0.87. In total, two statistically significant predictors of resilience scores were identified in the final model (see Table 4). The strongest predictors of resilience were the age of respondents (β = 0.385, *p* = 0.017) and education level (β = 0.320, *p* = 0.027). Therefore, those of an older age and educated to degree level and above scored higher on the resilience scale. The final model explained 16.8% of the variance in resilience scores.

### 3.5. Scale Factors

In order to understand which of the five factors identified in the original psychometric testing of the scale [31] were associated with age and education, Pearson’s product–moment correlations were performed on all five factors. The relationship between age and the factor representing positive acceptance of change and secure relationships demonstrated a moderate, positive correlation between the two variables (0.282, n = 56, *p* = 0.035). The results indicate that older age is associated with a more positive acceptance of change and secure relationships. There was no significant association with the other factors and age.

To test the hypothesis that there is a significant difference in the factor scores between those who have studied to degree level and those who have not, independent sample *t*-tests were conducted. There was a significant difference in scores relating to the ‘emotional and cognitive control under pressure’ factor (t(54) = 2.29, *p* = 0.013) among those having studied to degree level and above (M = 29.41, SD = 3.35) and those having studied to certificate and diploma level (M = 27.38, SD = 3.21). The magnitude in the means was moderate (Cohen’s d = 0.62).

## 4. Discussion

This pilot study aimed to explore the resilience of a sample of care home nurses. Overall, a moderate level of resilience was reported in the CD-RISC, with a mean score of 77.36 (SD = 11.31, range 52–100) in this pilot study of care home nurses. This is lower than the mean score of 80.7 reported by the CD-RISC scale authors in their USA-population-based study [31]. While we could not find the reported mean score for populations in the United Kingdom, the variability in CD-RISC scores across populations has been demonstrated, with lower mean resilience scores reported in a population-based study in Hong Kong (n = 10,997) (M = 59.99, SD = 13.92) [76]. The score in this pilot study is significantly higher than the mean score of 68.5 (19) recorded in a study, which examined medical doctors’ (n = 290) resilience in Northern Ireland [77]. Resilience in our study was measured during an ongoing pandemic, demonstrating a key characteristic of resilience, which is the presence of adversity [29]. It mirrors the findings of reviews of resilience in health care workers during the COVID-19 pandemic demonstrating a moderate level of resilience [51,78]. The protective nature of resilience was demonstrated in the inverse relationship between resilience and burnout and COVID-19-related anxiety [79]. One study of aged care nurses in the USA included in the review suggested that, similar to other groups of health care workers, social support and good leadership were central to resilience building and maintenance [51,79]. Interventions strengthening the defences and focusing on nurturing the protective attributes demonstrated during challenges are advocated [79]. Leaders in care homes should consider resilience building as an essential capability and a priority for the individual, the organisation and the sector. While a bottom-up approach to ownership and development of resilience interventions in care homes is advocated [3], a ‘buy in’ [80] is required by the organisation, where they see resilience as a core capability and strategy to promote retention. This bottom-up and top-down approach will allow for strategies enhancing resilience to become embedded in care home life.

A moderate level of resilience has been reported in studies of hospital nurses in China, Australia and the USA [41,81,82,83]. Other studies have reported lower resilience scores, such as a feasibility study among hospital nurses in one UK hospital [84] reporting a baseline mean resilience score of 67.6 (SD = 8.8) in the CD-RISC. A quasi-experimental study measuring the impact of a resilience training programme among critical care nurses in Iran also reported a lower pre-intervention CD-RISC mean score of 68 (SD = 7.5) [85]. However, in a recent review of studies investigating evaluated resilience interventions to foster nurse resilience, no study was found which looked specifically at care home nurses [61]. In recent reviews of the evidence exploring nurse resilience, positive thinking, maintaining a work–life balance, pride in the profession and planning for a better future were strategies that nurses employed, which not only helped overcome adversity but enabled learning and growth from their experiences [54,86]. However, it is unclear whether resilience results from positive adaptation at times of adversity, or whether the capacity to adapt in a positive manner precedes the adversity and is either innate or learned [29,80]. It may be a mixture of both; interventions supporting nurse self-esteem are needed, as these have been shown to influence adaptive coping strategies [87].

### 4.1. Aspects of Resilience at the Interpersonal and Intrapersonal Level

A possible suggestion for the moderate score on the CD-RISC—despite the prolonged challenges of the pandemic—is that significant challenges relating to systems, resources and support existed before the pandemic [88]. It could also be the case that the pandemic may have released an ‘untapped capacity’ within this workforce [19,89], allowing assets to emerge, which had not been required previously [6]. A sense of personal achievement and making a difference has been associated with feelings of resilience in a qualitative enquiry [81]. Indeed, an integrative review of the stress and resilience in the Australian workforce [54] highlighted self-efficacy mechanisms, such as self-reliance, positive thinking, passion and emotional intelligence, as important influences on resilience. It may be the case that the uniquely autonomous role of care home nurses and their contribution to a functioning health system have entered the public consciousness [2], enhancing the feelings of value and respect. Employing a salutogenic approach [90] to supporting the resilience of care home nurses enables a focus on assets, which individuals can bring, which can then be modelled, developed and nurtured. This might assist in dispelling the internalised negativity associated with resilience, which was found in in-depth qualitative interviews held with nurses reflecting on the impact of the COVID-19 pandemic [91]. This could potentially inhibit nurses from seeking help and advocating for resources to promote a healthier work environment [91].

At the intrapersonal level, age was found to have a positive relationship with resilience in this study, in keeping with other studies on resilience in hospital nurses [37,38,42,92]. Perhaps those who are older may have faced more life challenges and may have had opportunities to develop positive coping strategies [93]. Learning from experience has been reported as a central part of resilience in qualitative studies involving nurses who care for older people; it forms part of a protective ‘scaffolding’, which brings order and a sense of control [94,95]. Interestingly, in this study, a significant association was found between the factor explaining adaptation to change and secure relationships with age (*p* = 0.035). The influence on resilience at the interpersonal level of social networks and organisational support has been highlighted throughout the literature in reviews prior to and during the pandemic [51,54]. The strength garnered from within the organisation and supplemented by family and friends has been shown to be a key determinant of resilience [12,46,60]. These relationships may not feel as secure with younger nurses with less experience who may not have the confidence to tolerate change. In a synthesis of systematic reviews exploring the factors impacting resilience, a positive workplace culture, which enables a work–life balance, reducing isolation and promoting cohesion and collegiality were seen as significant factors in fostering resilience [80].

In a study of resilience in care home nurses using lived experience accounts, resilience was seen to be enforced by making a difference and the close relationship with residents [46]. Indeed, in a recent study of long-term care staff in Canada, the feeling of connection between staff and clients was highlighted as positively influencing resilience [96]. The impact of forced isolation and mask wearing has undoubtedly affected this potential for resilience building [8,95]. The nurse managers in this study scored lower on the CD-RISC than nurses who did not have a management role, almost reaching significance (*p* = 0.093). This differs from studies, which have shown those of higher clinical grade being more resilient [44]. One possible explanation may be that managers have had to implement protective policies and procedures, which have been unpopular with residents, communities and co-workers [7]. This may have distanced many managers from these essential sources of connection. Contrary to some previous studies [42] and in keeping with others [41], no significant association was found between marital status and resilience. Calling for a systemic change to improve the working conditions for nursing home leaders, a qualitative study undertaken in Canada found that nursing home leaders were at increased risk of burnout, challenged by inordinate workloads and mental distress [97].

The sample in this study was predominantly white, with resilience scores showing some mean differences but not at a significant level. In a large Canadian study of care assistants (n = 1194) working in nursing homes, not speaking English as a first language was associated with higher levels of emotional exhaustion and (*p* = 0.008) depersonalisation (*p* = 0.002). As resilience has been shown to act as a buffer to burnout and with a growing reliance on nurses from other countries in social care [19], a large-scale enquiry could inform the development of bespoke interventions and build on the assets, which a multi-cultural approach offers.

The importance of education as a potential predictor of resilience in this study correlates with the findings of a study of hospital nurses in Greece (n = 1012) [98]. Indeed, postgraduate education was the only socio-demographic characteristic, which predicted resilience in a hierarchical regression of multiple variables, including gender, age, marital status and religiousness, in a recent study of Taiwanese nurses (n = 813) (*p* = 0.003) [99]. Promoting a culture of personal and professional growth—with an emphasis on problem solving, building emotional capacity, dealing with uncertainty and self-efficacy—has been suggested as a means of enhancing resilience and creating a more stable workforce [54,100]. While nurses were the primary focus of this pilot study, it is recognised that the majority of care given to residents in care homes is provided by staff who are not registered nurses [3]. Exploring the resilience of these groups is important. Creating a profile of resilience among all staff in the care home sector will give some insight into particular groups who may need tailored support.

### 4.2. Strengths and Limitations

This exploratory pilot study is limited by a small convenient sample and cross-sectional design. The study was undertaken during the COVID-19 pandemic, a particularly challenging time for care homes. However, the findings give a first snapshot of the resilience of a sample of care home nurses in Northern Ireland, providing a novel insight into the levels of resilience and potential influencing factors. The study provides a valuable direction to inform the content and recruitment for future adequately powered studies, which aim to engage a broader population of participants who work in the social care sector. The sample is weighted towards white nurses with more experience. With increasing international recruitment within the sector, future studies should ensure that the voices of care home staff from other ethnicities and those with less experience are included. Future studies should include qualitative measures, such as interviews and focus groups, to enhance and develop the quantitative findings and provide a comprehensive picture of how the construct of resilience relates to care home nurses.

## 5. Conclusions

The findings of this pilot study resonate with some other studies of nurse resilience and not with others. One possible explanation for this inconsistency may be that adaptation in times of significant adversity is not linear, with many influences contributing to both positive and negative adaptation [101]. The socio-ecological model offers some insight regarding the influences on individual resilience, highlighting the dynamic cultural and context-based nature of the construct of resilience. The care home environment is uniquely placed within communities as part of an organisation and subject to the rules and regulations of both private and statutory organisations. However, care homes are heterogenous in terms of the location, size, organisation, age and the experience of care home staff [2]. Determining what makes care home nurses resilient requires a study of individual contexts and may be informed in future research by a realist approach to discover what factors impact resilience and in what context. A culture, which fosters resilience, requires an all-systems approach, which promotes individual assets, is enabled by organisations and supported at the interpersonal level by peers, friends and families, and affirmed by the community. Complementing larger scale quantitative data with qualitative enquiry may allow different facets of the construct of resilience to be explored, increasing our understanding of resilience and developing a framework, which promotes resilience across individual care home organisations. This will help capitalise on the strengths of the workforce, enabling and nurturing positive emotions and skills, which may act not only as ‘shock absorbers’ [30,102] but as drivers in promoting positive change and cohesion within a challenged health and social care system.

## Figures and Tables

**Table 1 healthcare-11-03120-t001:** Demographic characteristics of the sample of care home nurses.

Variable	Frequency	Percentage %
Gender		
Female	50	89%
Male	6	11%
Ethnicity		
White	48	86%
Filipino	4	7%
Black African	4	7%
Age		
≤39	14	25%
40–50	16	29%
51–55	12	21%
56+	14	25%
Education Level		
Certificate/Diploma in Nursing	24	43%
Bachelor’s Degree	22	39%
Postgraduate Certificate/Diploma	6	11%
Master’s Degree	3	5%
PhD	1	2%
Primary Role within the Home		
Nurse/Sister	19	34%
Manager	37	66%
Years of Experience		
≤11	17	30%
12–25	12	21%
26–32	15	27%
33+	12	21%
Dependants		
Yes	36	64%
No	20	36%
Marital Status		
Married/Domestic Partnership	45	80%
Single/Divorced	11	20%

**Table 2 healthcare-11-03120-t002:** Demographic characteristics with mean and standard deviation scores on the CD-RISC.

Demographic Variable	Frequency	Mean CD-RISC Score	Standard Deviation (SD)
Gender			
Male	6	77.4	11.9
Female	50	77.3	11.4
Ethnicity			
White	48	77.0	10.4
Filipino	4	76.8	
Black African	4	82.5	
Age			
≤39	14	74.6	8.8
40–50	16	77.2	13.0
51–55	12	76.8	10.8
56+	14	80.1	12.3
Marital Status			
Single/Divorced/Widowed	11	80.5	12.5
Married/Partnership	45	76.6	11.0
Dependants			
Yes	36	77.9	11.4
No	20	76.3	11.4
Education			
BSc +	32	79.3	11.0
Certificate/Diploma	24	74.8	11.4

**Table 3 healthcare-11-03120-t003:** Nursing and care home characteristics with mean and standard deviation scores on the CD-RISC.

Nursing and Care Home Characteristics	Number Total n = 56	Mean CD-RISC Score	Standard Deviation (SD)
Role in Home			
Nurse/Sister	19	79.6	10.8
Manager	37	76.2	11.6
Years of Experience (Quartiles)			
≤11	17	77.6	9.4
12–25	12	77.9	14.0
26–32	15	75.0	13.0
33+	12	79.0	9.3
Working in Care Home since COVID-19			
<12 months	3	86.0	9.5
>12 months	53	76.9	11.3
Urban/Rural			
Urban	42	77.5	11.2
Rural	14	76.9	11.9
Capacity of Home (Beds)			
1–30	12	76.1	10.2
31–50	22	77.4	10.1
51–80	17	78.2	14.7
81+	5	77.2	8.1
Shift Work			
Day	46	77.2	11.7
Night and Day	8	67.5	8.7
Nights	2	81.0	2.1
Employment Status			
Full Time	53	77.4	11.6
Part Time	3	76.0	6.6

**Table 4 healthcare-11-03120-t004:** Parameter estimates for the regression model with resilience as the dependent variable.

Predictor Variable *p* < 0.05	B Unstandardised Coefficient	Coefficients’ SE	Standardised Coefficient β	t	Sig.
Constant	84.99	9.43		9.02	<0.001
Age	0.46	0.19	0.385	2.52	0.017
Education	7.26	3.20	0.320	2.28	0.027
Nurse/Manager	5.81	3.40	0.245	1.17	0.093
Married/Single	5.03	3.68	0.178	1.37	0.178

## Data Availability

The dataset pertaining to the study can be found on the Queen’s University Belfast Research Portal. DOI. https://pure.qub.ac.uk/en/datasets/dataset-for-resilience-and-the-care-home-nurse-a-cross-sectional.

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
