# Peer review of "Exploring Resilience in Care Home Nurses: An Online Survey"

_healthcare, 2023, doi:10.3390/healthcare11243120_

Round 1

Reviewer 1 Report

Comments and Suggestions for Authors

Thank you very much for give me the opportunity to review this interesting paper entitled Exploring Resilience in Care Home Nurses: a pilot study Achievement. Altogether it is an interesting and quite well written pilot study that aimed to investigate the levels of resilience in care home nurses in Northern Ireland and to explore factors that may predict higher resilience in this cohort. As authors noted, the pandemic has exacerbated levels of psychological distress among health care workers, with nurses identified as a particularly at-risk group. However, there is a dearth of evidence as to which socio-demographic factors influence the resilience of care home nurses.

Say this, here are a few comments:

1.     Maybe you could highlight the purpose of your study more clearly in the abstract.

2.     Although the introduction is well written, offering definitions of the constructs that the article works with, as well as a good review of the state of the art, in my humble opinion the objective should be better outlined (with what is pointed out in line 133 it is not entirely clear whether the sample is only composed of home care nurses or also of social workers) and the hypotheses as well. Although this is a pilot study, in your introduction you provide results from previous studies that show the direction of the relationship between socio-demographic variables and resilience.

3.     I am not very sure of you organization in headings (you use method instead of material and methods) and subheadings. For example, 2.1. Research setting and sample, I am more used to the expression participants, and to the fact that even minimally some data will be offered that describes them in that section (although I understand that this broader description is made in the results section), and perhaps to that the characteristics of the study and its nature (or design) will be described in the procedure section.

4.     Please check the wording in line 183, it is not entirely clear whether you used a measure with 38 items or 25 items to assess resilience. Explain more clearly your measures and what demographics, etc., those other 14 items assessed.

5.     Perhaps you could restructure the presentation of your results by providing a short paragraph to help the reader follow you. First you seem to offer descriptive, then differential, then regression results, but the correlational results (which are usually presented before the regression analyses) are offered later.

6.     Although I understand that this is a pilot study, it might be interesting for you to refer to the effect size with the sample participating in it, and its adequacy from the point of view of some of the analyses you have carried out. In a similar vein, how did you transform the qualitative variables to incorporate them into a linear regression analysis?

7.     I am also more used to the fact that when the p is significant it is written with italics (p)

8.     Maybe the discussion section might be improved, in example, starting with remembering your goal and adding some more arguments regarding the practical implications and suggestions for future research of your work.

9.     Please check the format (for example lines 29, 40, 42, etc., it seems there are extra spaces, in line 95 it seems p is lacking (you use just the expression <0.001, in line 259 what does it mean @?)

10.  Please review the references, both in the text and in the references section, as they do not seem at all to follow the format indicated by the journal (you use parentheses () instead of brackets [ ]. Also, if possible, it would be interesting to incorporate a greater number of references, especially more recent ones (none of them dates from 2023).

Author Response

Dear Reviewer 1

Thank you so much for taking the time to review our manuscript. Your input has helped strengthen this paper. Below we have addressed each point. In the manuscript these are highlighted through track changes in the main manuscript

Reviewer 1 comment

Author Response

1.    Maybe you could highlight the purpose of your study more clearly in the abstract.

Thank you. We have now included in the abstract - To address this gap an online survey was undertaken to explore levels of resilience and potential influencing factors in a sample of care home nurses

2.    Although the introduction is well written, offering definitions of the constructs that the article works with, as well as a good review of the state of the art, in my humble opinion the objective should be better outlined (with what is pointed out in line 133 it is not entirely clear whether the sample is only composed of home care nurses or also of social workers) and the hypotheses as well. Although this is a pilot study, in your introduction you provide results from previous studies that show the direction of the relationship between socio-demographic variables and resilience.

Thank you for pointing out how this may be confusing. We have now included;

This study sought to determine levels of resilience in care home nurses in one region of the United Kingdom and to explore factors that may predict higher resilience. We have removed the reference to health and social care workers.

The results from previous studies are inconclusive regarding sociodemographic variables – we have highlighted this in the introduction

3.       I am not very sure of you organization in headings (you use method instead of material and methods) and subheadings. For example, 2.1. Research setting and sample, I am more used to the expression participants, and to the fact that even minimally some data will be offered that describes them in that section (although I understand that this broader description is made in the results section), and perhaps to that the characteristics of the study and its nature (or design) will be described in the procedure section.

Thank you we have changed this to Materials and Methods

We used subheadings for ease of reading. I have now labelled these 2.1., 2.2 using track changes. The term participant is used throughout, only when describing sampling do we use the term sample 

4.     Please check the wording in line 183, it is not entirely clear whether you used a measure with 38 items or 25 items to assess resilience. Explain more clearly your measures and what demographics, etc., those other 14 items assessed.

Thank you we have changed the text to reflect the content of the survey more succinctly

This study followed a quantitative, descriptive approach using a cross-sectional survey design to gather data. The survey included a 25-item resilience scale the Connor Davidson Resilience Scale (CD-RISC-25) Fourteen further items relating to demographics nursing role and characteristics of the care home identified through the literature as potentially influencing resilience levels, were also collected, In the data analysis section, the exact 14 items are listed

5.     Perhaps you could restructure the presentation of your results by providing a short paragraph to help the reader follow you. First you seem to offer descriptive, then differential, then regression results, but the correlational results (which are usually presented before the regression analyses) are offered later.

Thank you as per reviewer 4 I have referred in the data analysis section to the scale factors. This guided further inferential testing relating to which individual factors (see section 2.3) could be associated with the predictor variables.

 The procedure is explained in the data analysis section; however, the paragraph below could also be included.

In this section we describe the characteristics of the population and identify the mean resilience scores and significance across the participant demographics. Linear regression is used to identify potential predictors of resilience. Inferential testing on the scale factors is undertaken to explore associations.

6.    Although I understand that this is a pilot study, it might be interesting for you to refer to the effect size with the sample participating in it, and its adequacy from the point of view of some of the analyses you have carried out. In a similar vein, how did you transform the qualitative variables to incorporate them into a linear regression analysis?

Thank you. The analysis in this pilot was an exploratory analysis with the limitations of small sample size recognised. The effect sizes are reported for the scale factors. 

Some differences in scores were apparent on descriptive analysis such as higher scores for those who were older and those who were single. Participants who had dependants also scored higher. However comparing means via independent t tests and ANOVA reported no differences that were significant at the p ≤ 0.05 level

7.      I am also more used to the fact that when the p is significant it is written with italics (p)

Thank you we have changed this throughout the text

8.    Maybe the discussion section might be improved, in example, starting with remembering your goal and adding some more arguments regarding the practical implications and suggestions for future research of your work.

Thank you we have started with reiterating the aim of the study at each level we have made recommendations for practice and future research

9.    Please check the format (for example lines 29, 40, 42, etc., it seems there are extra spaces, in line 95 it seems p is lacking (you use just the expression <0.001, in line 259 what does it mean @?)

Spaces removed. Thank you for picking this up

These differences did not reach significance at the p ≤ 0.05 level.

(p <0.001) line 100

10. Please review the references, both in the text and in the references section, as they do not seem at all to follow the format indicated by the journal (you use parentheses () instead of brackets [ ]. Also, if possible, it would be interesting to incorporate a greater number of references, especially more recent ones (none of them dates from 2023).

Thank you

Square brackets have been included and some more recent references

91 Conolly A, Abrams R, Rowland E, Harris R, Couper K, Kelly D, et al. “What Is the Matter With Me?” or a “Badge of Honor”: Nurses’ Constructions of Resilience During Covid-19. Global qualitative nursing research. 2022;9:23333936221094862.

96.         Hung L, Yang SC, Guo E, Sakamoto M, Mann J, Dunn S, et al. Staff experience of a Canadian long-term care home during a COVID-19 outbreak: a qualitative study. BMC Nursing. 2022;21(1).

97.         Savage A, Young S, Titley HK, Thorne TE, Spiers J, Estabrooks CA. This was my Crimean War: COVID-19 experiences of nursing home leaders. Journal of the American Medical Directors Association. 2022;23(11):1827-32.

Reviewer 2 Report

Comments and Suggestions for Authors

I have received a manuscript titled “Exploring Resilience in Care Home Nurses: a pilot study”. It is a well-developed manuscript. However, the following points will help improve the manuscript;

1.       There is a need to clearly write the aim of the study in the abstract section

2.       There is no need to report the mean values in the methodology and results sections of the abstract.

3.       Similarly, the beta values and the values of significance are not generally reported in the abstract section.

4.       The in-text references must be provided in the square brackets instead of the round brackets. See for example – [1], [2,3,4] and so forth.

5.       There is a need to report the findings of the earlier studies viz-a-viz the gaps they have not covered for doing this study.

6.       While reporting the literature, there is no need to write the values of significance reported by the earlier studies.

7.       The methodology section must report the sampling technique, sampling procedure adopted and the data collection procedure used for the study.

8.       Results are acceptable

9.       Discussion is acceptable

10.   Instead of strengths and limitations the authors can provide the limitations and future direction heading and explain it.

11.   English language is fine.

12.   I have not checked the similarity and it stands the responsibility of the authors to check the plagiarism.

13.   Minor Revisions

Comments on the Quality of English Language

I have received a manuscript titled “Exploring Resilience in Care Home Nurses: a pilot study”. It is a well-developed manuscript. However, the following points will help improve the manuscript;

1.       There is a need to clearly write the aim of the study in the abstract section

2.       There is no need to report the mean values in the methodology and results sections of the abstract.

3.       Similarly, the beta values and the values of significance are not generally reported in the abstract section.

4.       The in-text references must be provided in the square brackets instead of the round brackets. See for example – [1], [2,3,4] and so forth.

5.       There is a need to report the findings of the earlier studies viz-a-viz the gaps they have not covered for doing this study.

6.       While reporting the literature, there is no need to write the values of significance reported by the earlier studies.

7.       The methodology section must report the sampling technique, sampling procedure adopted and the data collection procedure used for the study.

8.       Results are acceptable

9.       Discussion is acceptable

10.   Instead of strengths and limitations the authors can provide the limitations and future direction heading and explain it.

11.   English language is fine.

12.   I have not checked the similarity and it stands the responsibility of the authors to check the plagiarism.

13.   Minor Revisions

Author Response

Dear Reviewer, 2

Thank you so much for taking the time to review our manuscript. Your input has helped strengthen this paper. Below we have addressed each point. In the manuscript these are highlighted through track changes in the main manuscript

There is a need to clearly write the aim of the study in the abstract section

Thank you we have included this in the abstract - To address this gap an online survey was undertaken to explore levels of resilience and potential influencing factors in a sample of care home nurses in Northern Ireland between January and April 2022.

There is no need to report the mean values and Beta values in the methodology and results sections of the abstract.

These values have been removed

The in-text references must be provided in the square brackets instead of the round brackets. See for example – [1], [2,3,4] and so forth.

Thank you, this has been addressed throughout

There is a need to report the findings of the earlier studies viz-a-viz the gaps they have not covered for doing this study.

This has been reported in the introduction, Yet little is known as to the levels of resilience of the care home nurse, what unique attributes they bring to the work, what can be nurtured, supported and what can be learned. While there has been more focus on nurse resilience in recent years with studies involving hospital nurses in general and specialist units (1-4) there remains a dearth of research on supporting resilience of the care home nurse (5).

Reviews of interventions aiming to support nurse resilience highlight three main points: 1) the need for an unequivocal understanding of resilience, 2) that resilience should be measured as a primary outcome and 3) investigation of resilience should extend beyond the general hospital to workers in social care (6-9). T

 Instead of strengths and limitations the authors can provide the limitations and future direction heading and explain it.

The methodology section must report the sampling technique, sampling procedure adopted and the data collection procedure used for the study

Thank you we have included text below to emphasise further how the content and conduct of this study can inform further studies.

The findings give a first snapshot of the resilience of a sample of care home nurses in Northern Ireland providing a novel insight into levels of resilience and potential influencing factors. It provides valuable direction to inform the content and recruitment for future adequately powered studies that aim to engage a broader population of participants who work in the social care sector

This is included under research setting and sample (section 2.1.)

The sample and inclusion criteria for this pilot study were 1) registered nurses, and 2) presently employed in care homes in Northern Ireland. A convenience sample of nurses was recruited using Care Home support groups and professional networks of the research team. Through these networks, the managers of the care homes were contacted via an invitation email containing a link to the participant information sheet and survey and asked to share the survey with registered nurses working in their care homes. Consent to participate was gathered prior to accessing the questions on the survey. 

Reviewer 3 Report

Comments and Suggestions for Authors

It is an interesting topic and the text is well-written; I have two comments for improvement of the paper:

1. Could you please assess whether the people excluded may be different from those included? 

2. Could you please take into account statistically the potential correlation as a result of the clustering of nurses within care homes (for example, to determine if there is any correlation between variables within units/care homes)?  Or at least (another option), could you please mention that there may be potential correlation and future research should explore this further?  People in one care home may be different from those in another care home, and therefore, statistically, this should be taken into account - if you could include a segment on this in the discussion, that would be good.

Author Response

Dear Reviewer, 3

Thank you so much for taking the time to review our manuscript. Your input has helped strengthen this paper. Below we have addressed each point. In the manuscript these are highlighted through track changes in the main manuscript

1. Could you please assess whether the people excluded may be different from those included? 

The inclusion criteria were being a registered nurse working in a care home. The 16 cases that were removed had insufficient data to undertake any analysis. This study will inform the content and conduct of a larger study that will include social care workers from across different settings.

2. Could you please take into account statistically the potential correlation as a result of the clustering of nurses within care homes (for example, to determine if there is any correlation between variables within units/care homes)?Or at least (another option), could you please mention that there may be potential correlation and future research should explore this further?  People in one care home may be different from those in another care home, and therefore, statistically, this should be taken into account - if you could include a segment on this in the discussion, that would be good.

Thank you for this suggestion. We have amended the conclusion to highlight the importance of context and the diversity of care homes

‘However, care homes are heterogenous in terms of location, size organisation and age, and experience of the care home staff. (10). Determining what makes the resilient care home nurse requires study of individual contexts and may be informed in future research by a realist approach to discover what factors impact on resilience and in what context.’

Reviewer 4 Report

Comments and Suggestions for Authors

Thanks for the opportunity to review this manuscript!

Author Response

Dear Reviewer, 4

Thank you so much for taking the time to review our manuscript. Your input has helped strengthen this paper. Below we have addressed each point. In the manuscript these are highlighted through track changes.

First of all, the mention in the title that the work is a pilot study. This mention calls into question the validity of the study results. In the end, the study is done on a small sample (56), but it can be considered for what it offers.

I suggest that the authors reflect on this aspect.

Thank you for this suggestion. As a team this has stimulated quite a bit of debate! As indicated in the manuscript there is a gap in the literature relating to resilience outside of the hospital setting. We have used this study to inform a larger study of resilience in a broader population of workers in social care settings.

However, we believe the findings can stand alone and while the sample is small there are valid findings from this exploratory study.

We have removed the term pilot from the  title:

Exploring the resilience of care home nurses_ an online survey

Line 232 “In total 72 responses were received with 56 valid datasets.” - The presentation of the initial number of respondents in the study does not seem relevant to the study.

Thank you

We feel that while the data collected on the 16 participants was minimal it is still collected data.

Lines 272-285 “Scale Factors” -  In the study, analyzes and results that include the 5 factors are performed and presented, but this is not mentioned in “Data Analysis” as well as in “Discussion” sections.

Also, another aspect that needs to be       reviewed is the presence of the name "nurses" in the description of the participants' profession, considering that only a part of them graduated from a specialized school and received a nursing diploma.

I think that the study can be published after a serious revision, emphasizing the limit represented by the sample size.

Thank you for this. We have amended this in the data analysis section

The results of the linear regression guided further inferential testing relating to which individual factors (see section 2.3) could be associated with the predictor variables.

In the discussion the potential associations are discussed -  Interestingly in this study a significant association was found between the factor explaining adaptation to change and secure relationships with age (p=0.035).

Our participants were all registered nurses as per the inclusion criteria. Some of the nurses registered many years ago before Diplomas and Degrees were required for nursing. They would all have undergone training at a specialised school of nursing.

Thank you we have revised the limitations section to emphasise the limitations of the small sample.

This exploratory pilot study is limited by using a small convenient sample and cross-sectional design. The study was undertaken during the COVID – 19 pandemic, a particularly challenging time for care homes.  However, the findings give a first snapshot of the resilience of a sample of care home nurses in Northern Ireland providing a novel insight into levels of resilience and potential influencing factors. It provides valuable direction to inform the content and recruitment for future adequately powered studies that aim to engage a broader population of participants who work in the social care sector.   The sample is weighted towards white nurses with more experience.

Round 2

Reviewer 1 Report

Comments and Suggestions for Authors

Thank you for giving me the opportunity to review this article again. I think the authors have improved it considerably, although it is important that they still pay attention to some formatting issues such as line 44, 97-98 and including bibliographic references from 2023.